# Evaluation of Anti-Obesity and Antidiabetic Activities of *Orostachys japonicus* in Cell and Animal Models

**DOI:** 10.3390/ph17030357

**Published:** 2024-03-10

**Authors:** Ramakanta Lamichhane, Prakash Raj Pandeya, Kyung-Hee Lee, Gopal Lamichhane, Jae-Young Cheon, Hyo Shin Park, Nguyen Quoc Tuan, Hyun-Ju Jung

**Affiliations:** 1Department of Pharmacy, Kathmandu University, Dhulikhel 45200, Nepal; ramakanta.lamichhane@ku.edu.np; 2Department of Oriental Pharmac and Wonkwang-Oriental Medicines Research Institute, Wonkwang University, Sinyong-dong, Iksan 570-749, Republic of Korea; fuhaha112@naver.com (K.-H.L.); lamichhanegopal1@gmail.com (G.L.); cjy21750@naver.com (J.-Y.C.); phs4846@naver.com (H.S.P.); quoctuan301281@gmail.com (N.Q.T.); 3Department of Animal and Food Sciences, University of Kentucky, Lexington, KY 40546, USA; pandeya.praj@gmail.com; 4Department of Nutritional Sciences, Oklahoma State University, Stillwater, OK 74078, USA

**Keywords:** *Orostachys japonicus*, anti-obesity, antidiabetic, 3T3-L1, high-fat, high sugar diet

## Abstract

*Orostachys japonicus* is a popular traditional medicinal herb used in Asian countries. This study is focused on evaluating its role in lipid and glucose metabolism in cell and animal models to establish the plant as an anti-obesity and antidiabetic herb. A butanol fraction of *O. japonicus* was used in the study. The lipid production was evaluated by the Oil Red O technique while the expression of adipogenic markers by Western blotting and RT-PCR using 3T3-L1 preadipocyte. The effect on glucose uptake activity was evaluated in C2C12 myoblast cells. The animal study was carried out in C57BL mice to evaluate anti-obesity activity using the high-fat diet model. The evaluation of serum lipid, blood glucose, adipogenic and fibrosis markers in the liver, and fat deposition in the liver and adipose tissue (by histology) of mice was conducted. Butanol fraction of *O. japonicus* significantly inhibited the lipid production in the 3T3-L1 cells and reduced the expression of PPARγ, C/EBPα, SREBP-1c and aP2. It enhanced glucose uptake in insulin-resistant C2C12 myoblast cells. It reduced body weight, triglycerides, and blood glucose in the obese mice. It significantly inhibited lipid accumulation in the liver and adipose tissue of obese mice along with suppression of expression of adipogenic and fibrosis markers in the liver. In summary, supporting the previous results, this study helped to establish the potent anti-obesity, antidiabetic, and liver-protecting effect of the butanol fraction of *O. japonicus*.

## 1. Introduction

Obesity and diabetes are chronic diseases that may lead to long-term disability, hospitalization, reduced quality of life and even death [1]. Both are the risk factors for various chronic conditions including hypertension, high cholesterol, stroke, heart disease, kidney disease, amputation, certain cancers and arthritis [2]. Obesity is a metabolic disorder caused mainly due to the imbalance in food intake and energy expenditure. It results in abnormal or excessive cholesterol in blood and fat accumulation in tissue. It is one of the main public health problems worldwide because its prevalence is constantly and worryingly increasing [3]. On the basis of the facts available, it has been estimated that in the USA, more than 85% of adults will suffer from overweight or obesity by 2030 [4]. With the increasing incidence of obesity, the prevalence of diabetes (type 2) is also increasing day by day. It is because people with obese conditions are at an increased risk of developing type 2 diabetes. More than 85 percent of people who have type 2 diabetes are overweight, and more than 50 percent are obese [5]. Type 2 diabetes is also a metabolic disorder, characterized by chronic hyperglycemia with disturbances of carbohydrate, fat and protein metabolism resulting from insufficient insulin secretion, insulin resistance or both. Due to modernization and sedentary lifestyle, the incidence of obesity and diabetes is rising continuously.

The treatment of obesity and diabetes includes various strategies and, among them, lifestyle intervention (diet and exercise) is the first priority. Next is the pharmaceutical intervention as there are various categories of prescription medications available [6]. These modern allopathic medications are not free from the side effects and complications that have restricted their uses and are sometimes banned in many countries. So, the use of traditional herbs and herbal products as alternative therapeutic medications for the treatment of obesity and diabetes has been increasing. The yearly increase in the market value of dietary supplements and herbal medicines indicates the attraction of people towards herbal treatments besides allopathic treatment [7,8]. This ultimately has increased the research work in herbs and phytochemicals. The potential anti-hyperlipidemic and antidiabetic effects of several traditional plants have been reported in various studies [9,10,11,12].

*Orostachys japonicus* is a perennial herb that grows in mountainous regions, belonging to the family Crassulaceae and native to East Asia. The growth habit of the plant is similar to the pine cone resembling pine leaves or flowers. It is a succulent plant with a rosette appearance. It contains many bioactive compounds such as organic acid, sugars, triterpenoids (glutinol, glutinone, friedelin, epi-friedelanol, β-amyrin, and taraxerone), triterpenes, phytosterols (β-sitosterol and campesterol), sterol glucosides, and flavonoids. Epicatechin gallate, quercetin, and kaempferol are known to be the major flavonoids found in the plant [13,14]. It is a widely used herb in various traditional medicine systems of China, Korea, and other Asian countries, and has been reported in various research articles for its promising biological activities [15]. Traditionally, the plant has been used for fever, inflammation, bleeding, intoxication, and hemostasis. Kim et al. studied on anti-hyperlipidemic effects of the flavonoid-rich fraction from the methanol extract of *O. japonicus* in rats [16]. A study conducted by Lee et al. showed its hypolipidemic and hypoglycemic effects in streptozotocin-induced diabetic rats [17]. Koppula et al. reported the anti-fibrotic effects of *O. japonicus* on hepatic stellate cells and thioacetamide-induced fibrosis in rats [18]. In this study, we aimed to establish the anti-obesity and antidiabetic activity of *O. japonicus* by including various studies in cell lines (3T3-L1 and C2C12) and animals, which were not conducted in previous studies. Butanol fraction of *O. japonicus* (BuOJ) was preferred in this study in contrast to major previous studies conducted on ethyl acetate fraction [16,19,20]. The use of butanol as an extracting solvent was expected to elute a high proportion of polar compounds like flavonoids which would show major biological activities.

## 2. Results

### 2.1. In Vitro α-Glucosidase Inhibition Assay

α-Glucosidase is one of the important carbohydrate-hydrolyzing enzymes, which plays a vital role in the breakdown of starch, glycogen and polysaccharides of our food into smaller monosaccharides. So, α-glucosidase is responsible for postprandial hyperglycemia, even though this is a normal process of metabolism. Hence, inhibitors of α-glucosidase would be potential antidiabetic medication to control hyperglycemia as they delay carbohydrate digestion, which consequently helps to maintain the glucose level in the normal range. The α-glucosidase inhibition activity of water extract of *O. japonicus* has been reported but there is no sufficient evidence to say how strong it is [21]. So, in this study, we evaluated the α-glucosidase inhibition activity of BuOJ and compared it with the standard pharmaceutical medication, Acarbose. The α-glucosidase inhibition activity of BuOJ was evaluated by its efficiency in inhibiting the enzymatic hydrolysis of the substrate p-nitrophenyl-α-D-glucopyranoside (p-NPG) to p-nitrophenol (monitoring at 405 nm) by the enzyme, α-glucosidase. The IC_50_ values of BuOJ and the positive control Acarbose are given in Table 1. The IC_50_ value of the fraction was lesser compared to the positive control. This suggests that *O. japonicus* has greater α-glucosidase inhibiting activity than that of Acarbose.

### 2.2. Cell Viability and Glucose Uptake Study

The data of the cell viability study (Figure 1A) revealed that the BuOJ was non-toxic to C2C12 cells up to the concentration of 120 µg/mL. The BuOJ was investigated to determine whether it would enhance the activity of insulin or not. The differentiated C2C12 myotubes were treated with insulin along with or without sample, i.e., butanol fraction (100 and 120 µg/mL). After starving the cells for 16 h in low glucose/serum-free media, the cells were treated with 2NBDG. The uptake of 2-NBDG was imaged using a fluorescence microscope (Figure 1B). Densitometric analysis of the images was carried out to evaluate the relative fluorescence intensity as shown in the bar diagram (Figure 1C). The results showed higher 2-NBDG fluorescence intensity (corresponding to glucose uptake) in combined treatment of sample and insulin compared to insulin-only treatment. The results confirmed that the supplementation of butanol fraction enhanced the activity of insulin. The effect of BuOJ on insulin-resistant C2C12 myoblasts was also evaluated by 2-NBDG uptake, as shown in Figure 1D. We observed a rise in glucose uptake in the insulin-treated group compared to the non-treated control. However, the addition of dexamethasone to the insulin-treated cells showed a fall in glucose uptake, indicating the induction of insulin resistance due to dexamethasone. When the insulin-resistant C2C12 myoblast was treated with butanol fraction (80, 100 and 120 µg/mL), there was again a significant increase in glucose uptake (Figure 1D). This revealed the ameliorating effect of BuOJ against dexamethasone-induced insulin resistance. BuOJ increased the glucose uptake in a dose-dependent manner. The higher concentration 100 and 120 µg/mL showed a significant increase in glucose uptake. Interestingly, the fraction ameliorated the impairment of glucose uptake in the insulin-resistant C2C12 myoblasts. 

### 2.3. Inhibition of Lipid Production in 3T3-L1 Cells

In this study inhibition of lipid production by BuOJ in 3T3-L1 cells was evaluated. The results of cell viability (Figure 2(BI)) indicated that the BuOJ, below the concentration of 130 µg/mL, was safe for the 3T3-L1 preadipocytes. Three concentrations (60, 80 and 120 µg/mL) of BuOJ were used for the evaluation of the anti-adipogenic activity. The quantitative evaluation showed significant inhibition of lipid production (around 50%) after the treatment of the 80 and 120 µg/mL of BuOJ (Figure 2(BII)). The pictures of the ORO assay (Figure 2C), which were captured from the microscope, also showed the reduction in the lipid droplets (red) in the butanol fraction treated cells compared to the control.

### 2.4. Inhibition of Adipogenic-Markers in 3T3-L1 Cells

The production of lipid or lipogenesis in the adipose tissue starts with a process called adipogenesis, which leads to the differentiation of pre-adipocytes to adipocytes. Different types of markers play a role in the initiation of adipogenesis and lipid production. It is well established that PPARγ and C/EBPα are the important proteins for the initiation of lipogenesis. The expression of major adipogenic proteins PPARγ and C/EBPα were evaluated from the Western blotting analysis in 3T3-L1 cells differentiated with or without the presence of BuOJ. The fraction significantly inhibited the expression of PPARγ and C/EBPα proteins in a concentration-dependent manner (Figure 3A). At 120 µg/mL, the expression of PPARγ and C/EBPα proteins was reduced by 30–40%. From the QPCR analysis, the effect on the expression of mRNA of different adipogenic markers (PPARγ, C/EBPα, SREBP-1c, aP2, adiponectin and leptin) was evaluated in 3T3-L1 differentiated cells with or without the presence of BuOJ (Figure 3B). The fraction significantly reduced the expression of mRNA of PPARγ, SREBP-1c, aP2 and leptin in a concentration-dependent manner. PPARγ expression was reduced by more than 40%, while C/EBPα expression was reduced by more than 20%. The expression of SREBP-1c, aP2, and leptin was inhibited by more than 40%. Interestingly, the mRNA of adiponectin was found to be overexpressed by the fraction (Figure 3B).

### 2.5. Animal Experiment: Body Weight Gain, Food Intake and Organ Weight

The in vivo high-fat diet study in mice was conducted to evaluate the anti-obesity activity of the BuOJ. The average body weight of each group of mice was recorded at every nine-day interval throughout the experiment (Figure 4A). We can see a continuous increase in body weight (sharp or flat) in all groups of mice as the experiment proceeds. As presented in Table 2, the HFD group gained the maximum weight of 33.02 ± 1.9 g, followed by BF 100 (32.39 ± 1.2 g) and BF 200 (30.44 ± 2.3 g). The normal group showed a minimum weight gain of 27.20 ± 2.1 g. The results of total food intake, body weight change and FER are given in Table 2. The HFD group gained weight around 10 g, which was around 2.5 times greater than that of the normal group. BF 100 was not so effective in controlling the body weight as the weight gain of BF 100 was similar to the HFD group. However, BF 200 was effective in controlling HFD-induced body weight gain. BF showed a weight gain of 8.651 ± 0.5 g, which was significantly different from that of the HFD group. The food intake of the high-fat diet provided to groups (HFD, BF 100 and BF 200) was lesser than the normal group, which may be due to the high energy density of their diet compared to a normal diet. The high-fat diet eating groups showed higher FER which supports their increase in body weight due to high-calorie diet. However, among them, the BF 200 showed the lowest FER value which confirms that supplementation of BF 200 greatly suppressed the body weight despite the intake of a high-calorie diet. 

The weight of different organs (liver, spleen and kidney) was measured to evaluate any toxic effects after the administration of BuOJ. There was no significant change in the weight of organs of sample-fed groups compared to the normal groups (Figure 4B). This indicated that the butanol fraction at the doses of 100 and 200 mg/kg/day is not toxic to the mice.

### 2.6. Blood Glucose, Serum Lipid and Accumulation of Lipid in White Adipose Tissue and Liver 

The obese mice (HFD group) showed an increase in blood glucose levels compared to the normal group while supplementation of BF 200 was able to reduce the blood glucose level in mice (Figure 5A). Both doses of BF 100 and 200 samples were able to reduce the serum lipid of high-fat diet-fed mice (Figure 5B). This clearly indicates that BuOJ can be effective in controlling glucose and lipid levels in the blood.

The results of the weight of WAT in different groups of mice are given in Figure 5D. The significant increase in the weight of WAT in the HFD group compared to the normal group indicates the obese condition in the HFD group. The quantitative data showed a significant difference in the weight of adipose tissue between the HFD and BF 200 groups. As shown in the histological sections (Figure 5C), the small-sized adipocytes in normal groups have turned to larger-sized adipocytes (more than two times) in the HFD groups due to the excessive accumulation of lipids. Two doses of BuOJ were administered along with the high-fat diet and the greater dose, i.e., BF 200 showed stronger lipid inhibition action comparatively to that of BF 100. The observation from the histological sections revealed a great reduction in the size of adipocytes in BF 200 compared to the HFD group. The morphology of adipocytes in the BF 200 group was similar to the normal group (small-size adipocytes) conforming to the inhibition of lipid accumulation by BF 200 supplementation.

The histology of liver tissue also reveals the inhibition of fat deposition in the liver by BuOJ (Figure 5E). The HFD group showed a large number of lipid droplets, indicating the fatty liver. However, the oral dosing of BuOJ inhibited lipid accumulation in the liver. BF 200 showed greater activity against lipid accumulation in the liver compared to BF 100. 

### 2.7. Adipokine and Fibrosis Markers Expression in Liver 

aP2 and LPL are the major proteins for fat metabolism in liver cells. Excessive accumulation of fat results in chronic inflammatory reactions, leading to damage of the liver cells which results in fibrosis in the liver. So, with the increase in fat accumulation, the fibrosis markers TGFβ and IL-6 are found to increase. To study the fat deposition and fibrosis in the liver, the expression of mRNA of respective markers was evaluated in the liver tissue samples of each group of mice using the QPCR technique. As shown in the results (Figure 6), the liver cells of the HFD group showed excessive expression of mRNA of aP2, LPL, TGFβ and IL-6 compared to the normal group. This revealed the excessive fat deposition and fibrosis in the HFD group liver due to a high-fat diet. However, oral administration of BuOJ countered the effect of a high-fat diet on the liver. It also decreased the expression of mRNA of all those markers (an indicator of fat deposition and fibrosis). There was a decrease in the expression of mRNA of all the markers in a dose-dependent manner. This indicated the protective role of BuOJ during the time of obesity-induced inflammation and fibrosis in the liver.

## 3. Discussion

Obesity and diabetes are closely related problems since obesity is one of the major factors for developing diabetes in people. As both diseases are chronic in nature with several complications and comorbidities, they have a significant impact on the social, financial, and psychological status of the individuals. The prevalence of obesity and diabetes is increasing rapidly around the globe, claiming millions of lives every year [22,23]. Huge numbers of conventional (allopathic) medications are available for both diseases. However, the fundamental treatment is lifestyle modification (extreme dietary energy restriction and physical activity/exercise), as both diseases fall in the category of lifestyle disease [24]. When the fundamental treatment approaches fail to control the diseases, then pharmaceutical interventions are required. Different types of medications are available for obesity (orlistat, sibutramine, etc.) and diabetes (biguanides, sulfonylurea, thiazolidinedione, insulin, etc.) [22,23]. The difficulties in the availability of this medication, along with their harmful side effects, have resulted in a deviation of people’s interest and attraction towards plant and herb-based medications [25,26]. In addition, herbal medications are said to possess very few or no side effects. Many herbal medications are polyherbal preparations that have synergistic effects that potentiate the antidiabetic and anti-obesity activity [24]. Until now, many anti-obesity and antidiabetic herbs (medicinal plants) have been identified and thoroughly studied. Many of them have even been formulated into various dosage forms.

*Orostachys japonicus* is a widely used traditional Chinese herb [27]. The chemical constituents and pharmacological activities of *O. japonicus* have been studied extensively. Various in vivo studies reported the strong anti-obesity and antidiabetic activity of *O. japonicus*. However, anti-adipogenic and antidiabetic studies at the cellular level for *O. japonicus* are still lacking. A study reported the anti-adipogenic activity of *O. japonicus* in 3T3-L1 cells but the sample used was highly non-polar (dichloromethane fraction of *O*. *japonicus*) [28]. All the available in vivo studies for the anti-obesity and antidiabetic activity of *O. japonicus* have used polar fractions for the analysis. So, for the cellular-level study, we also used the polar fraction. This study further helped for proper correlation of the in vivo and in vitro results. In this study, BuOJ, a highly polar fraction was used to evaluate the anti-adipogenic and antidiabetic activity in cellular model. Polar fractions constitute important phytochemicals like polyphenols, flavonoids, terpenoids, glycosides, etc., which have various medicinal properties. So, for various human ailments, the polar fractions of plant extracts are commonly used.

The BuOJ showed significant glucose uptake in the insulin-resistant C2C12 cells. In this study, we induced insulin resistance in the C2C12 cells with dexamethasone. Various studies have shown the detrimental effect of dexamethasone in C2C12 cells leading to less sensitivity to insulin [29]. In our study, there was also a severe decrease in insulin-mediated glucose uptake after the dexamethasone treatment, indicating the achievement of an insulin-resistant state in C2C12 cells. However, increased glucose uptake after the addition of BuOJ indicated the insulin-resistant ameliorating effect of *O*. *japonicus*. This is the first type of study reported for *O. japonicus*. Type 2 diabetes is mainly associated with insulin resistance, which leads to decreased function of skeletal muscle. Skeletal muscle is the primary site for glucose absorption. It is responsible for more than 70% of insulin-mediated glucose uptake [30]. Insulin receptor substrates (IRS) are the cell receptors (in muscle) to initiate insulin-medicated glucose uptake. During the diabetic state (Type 2), these receptors are less sensitive to insulin and this pathological condition is called insulin resistance. Insulin resistance leads to a dramatic decrease in glucose uptake in muscle cells, which are the major glucose-storing cells.

The promising results of BuOJ are that being able to show α-glucosidase inhibition activity has also helped it to further potentiate the antidiabetic activity. This is a local action in the stomach, which reduces glucose absorption by inhibiting α-glucosidase, a prominent enzyme that initiates the preliminary metabolic breakdown of glycogen and other big and small polysaccharides to small glucose molecules. Inhibition of α-glucosidase ultimately plays a vital role in decreasing blood glucose levels by reducing the level of overall glucose absorption in the stomach.

As mentioned earlier, there are various articles that report the study of the anti-obesity activity of BuOJ in high-fat diet animal models. Such in vivo studies have revealed the anti-hyperlipidemic activity of different extracts (methanol, ethanol, water) and fractions (ethyl acetate, butanol) of *O. japonicus*. Our results also support the results of previous studies carried out in animal models. The accumulation of fat in adipose tissue and blood (triglycerides) due to high-fat diet feeding was able to normalize by the treatment of BuOJ. In addition, we also revealed its ameliorating role against liver steatosis from the results of liver histology. The anti-obesity study of the BuOJ using a high-fat diet animal model is being reported for the first time in our study. Among the two doses of BuOJ, the higher dose, 200 mg/Kg was highly effective in reducing the body weight of high-fat diet-fed mice compared to the control. The reduction in body weight was possible due to the inhibition of lipids in the adipose and liver by the BuOJ, as illustrated in the results of the weight of adipose tissue and histology. In addition to that, BuOJ also successfully reduced the level of lipids in blood. Thus, *O. japonicus* has established itself as a highly effective alternative for obesity treatment.

Adipose tissue, where fat accumulation occurs, is the important site for the storage depot of excessive energy supplied to the body [31]. During obesity, excessive growth of adipose tissue occurs through the hypertrophy and hyperplasia of adipocytes [31]. So, the inhibition or blocking of lipid synthesis in the adipocytes would be a major approach for anti-obesity treatment. The anti-adipogenic activity of BuOJ in 3T3-L1 was further studied at the molecular level to find the effect on proteins and genes for lipogenesis. The expressions of different adipogenic markers in protein and gene levels were evaluated. PPARγ and C/EBPα are the important regulators responsible for the initiation of adipogenesis [32,33,34,35,36]. It has been established that the first transcription factor induced by MDI in 3T3-L1 preadipocytes is C/EBPα [37]. C/EBPα will mediate the expression of PPARγ and work together in promoting adipogenesis [38,39]. From both Western blotting and QPCR, the expression of both C/EBPα and PPARγ was inhibited in both gene and protein levels by the BuOJ, which sharply reduced the lipid production lipid in the 3T3-L1 cells, as seen in the ORO results.

SREBP-1c has been identified as an important regulator of lipid production and fatty acid synthesis, as well as a supporter of the expression of PPARγ ligands [40]. aP2 or the adipose fatty acid binding protein is highly expressed in the adipogenesis process and plays an important role in obesity-induced insulin resistance [41,42]. aP2 is expressed in adipocytes at the terminal phase of cell differentiation [43]. It plays a vital role in the release and transport of fatty acids during lipogenesis in the adipocytes [44,45]. It has been reported that expression of leptin increases during adipocyte differentiation [46]. The inhibition of expression of mRNA of SREBP-1c, aP2 and leptin by the BuOJ in 3T3-L1 further supports its role in inhibition of lipid production. In the in vivo study, also, the expression of aP2 in the liver cells of BuOJ-fed mice was significantly lesser compared to the HFD-only fed group. So, in both cell and animal studies, the BuOJ was able to inhibit the aP2 expression and play a role in the reduction of lipid production.

Another important thing revealed from this study was that *O. japonicus* showed a protective role in the liver against the high-fat diet-induced fatty liver. We can see the higher production of inflammatory markers like IL-6, and TGFβ in obese mice. IL-6 is considered an inflammatory cytokine that increases hepatic insulin resistance [47]. Similarly, the increase in TGFβ increases fibrogenesis [48]. In our study, BuOJ has been able to reduce such inflammatory markers in the liver of high-fat diet-fed mice. The protective role of BuOJ in fatty liver is being reported for the first time. 

In different studies, it has been established that adiponectin promotes insulin sensitivity in the body [49,50]. So, as shown in our results, the increased expression of adiponectin by the BuOJ may help to reduce insulin resistance in diabetic conditions.

The findings in this study provide sufficient evidence that the plant *O. japonicus* can be utilized in herbal preparations for obesity, diabetes and liver fibrosis. It can be used in combination with or alone to prepare anti-obesity and antidiabetic herbal formulations. It can also be used in various nutraceuticals for controlling obesity, preventing diabetes and protecting the liver from fibrosis. 

## 4. Materials and Methods

### 4.1. Materials 

A 3T3-L1 cell line and a C2C12 (mouse myoblast) cell line were obtained from the American Type Culture Collection (Rockville, MD, USA). Dulbecco’s Modified Eagle’s Medium (DMEM; Hyclone, Logan, UT, USA), dexamethasone, and free fatty acid-free bovine serum albumin (FFA-free BSA) were purchased from Sigma (St. Louis, MO, USA) and 2-deoxy-2-[(7-nitro-2,1,3-benzoxadiazol-4-yl)amino]-D-glucose (2-NBDG) for glucose uptake experiments was obtained from Invitrogen Corporation (Carlsbad, CA, USA).

### 4.2. Extraction and Fractionation

Dried plants of *O. japonicus* were ordered from Herb Co. Limited, Seoul, Republic of Korea. It was extracted with ethanol using heat. The dried aerial parts of the plants were cut into pieces and packed in a round bottom flask. Ethanol, the extracting solvent, was added to the flask, the condenser was fitted and the solvent was heated to boiling for two hours. Extraction was repeated using a new solvent for the second and third cycles. The collective ethanol extract was filtered and dried in a rotary evaporator. The dried ethanol extract was suspended in water and fractionated with chloroform and then butanol to give chloroform, butanol and water fractions. The purpose of skipping the fractionation with ethyl acetate was to increase the proportion of polar compounds in the butanol fraction. The previous study has also shown a very low yield value of ethyl acetate fraction (6.4%) compared to the butanol fraction (20%) [20]. So, skipping the fractionation with ethyl acetate would also increase the yield of the butanol fraction. 

### 4.3. In Vitro α-Glucosidase Inhibition Assay

α-Glucosidase inhibitory assay was carried out according to the protocol mentioned in an article by McCue et al., with some modifications [51]. α-Glucosidase of 0.3 U/mL was prepared using 0.2 M phosphate buffer (pH 7.0). p-nitrophenyl glucopyranoside (pNPG) substrate of 0.3 mM was prepared in 0.2 M phosphate buffer (pH 7.0). The samples (in DMSO) were diluted with the buffer to prepare the required concentrations. In a microplate 20 μL of buffer solution, 30 μL of samples and 30 μL of enzyme solution were mixed and incubated for 15 min at 37 °C. After that, 30 μL of pNPG solution was added to each sample mix, control and blank and again incubated for 15 min at 37 °C. At last, 120 μL of 0.2 M sodium carbonate solution was added to stop the reaction. The BuOJ was maintained from 0.015 to 1 mg/mL (final concentration). In the blank, buffer solution was placed instead of the enzyme solution. The control consisted of a buffer solution. The absorbance was measured with a microplate reader at 405 nm. Acarbose was used as a positive control (10–8000 μM). The experiment was conducted in triplicate. The enzyme inhibitory rates were evaluated as follows:$$Inhibition\;\%=\frac{[A_{control}-\left(A_{sample}-A_{sample\;blank}\right)]\times 100}{A_{control}}$$

A_control_ = absorbance of control;

A_sample_ = absorbance of sample;

A_blank_ = absorbance of the blank.

The IC50 values of samples were calculated and reported as the mean ± standard deviation (SD) of three experiments.

### 4.4. Cell Viability 

Cell viability for the butanol fraction was evaluated by MTT (3-(4,5-Dimethylthiazol-2-yl)-2,5-diphenyltetrazolium bromide) assay. The 3T3-L1 pre-adipocyte cells were maintained in DMEM supplemented with 10% bovine. C2C12 muscle cells were cultured in high-glucose DMEM, supplemented with 10% FBS (fetal bovine serum). Media for both cells were supplemented with 1% antibiotic solution (penicillin, streptomycin) and the incubator was maintained at 37 °C with 5% CO_2_. Cells (of 0.5 × 10^4^ cells/100 μL in each well) were seeded in 96-well plates. After 100% confluence, the cells were treated with different concentrations of butanol fraction for 48 h and then MTT solution (final concentration 1 mg/mL) was added. After three hours, the formazan complex was dissolved with DMSO and the absorbance was measured in an ELISA reader.

### 4.5. Glucose Uptake Assay in Normal Myotubes

The glucose uptake was determined by evaluating the uptake of the fluorescent derivative of glucose called 2-NBDG in the C2C12 cells. DMEM high glucose (Hyclone, Logan, UT, USA) media containing 10% FBS and 1% penicillin–streptomycin was used as culture media. Cells were seeded in 96-well plates (black) at a density of 0.5 × 10^4^ cells/well (100 μL/well). When 80–90% cell confluency was attained, the media was changed to DMEM with 2% (*v*/*v*) horse serum (Hyclone, Logan, UT, USA) for inducing differentiation. Then, the cells were washed with warm PBS and treated with BuOJ at a concentration of 100 and 200 µg/mL prepared in Dulbecco’s Modified Eagle’s Medium Glucose-free medium (DMEM GF medium; Welgene, Daegu, Republic of Korea). After 4 h of sample treatment, the cells were treated with insulin (1 μM) for 30 min. Then, 2-NBDG solution (1 µM) was added to the plates for 2 h. The uptake of 2-NBDG was imaged using a fluorescence microscope.

### 4.6. Glucose Uptake Assay in Insulin-Resistance Induced Myotubes

The C2C12 cells were seeded in 96-well plates, as mentioned above, and cultured until 80% confluency was attained. Then, differentiation was induced by DMEM media with 2% (*v*/*v*) horse serum. The differentiated myotubes were exposed to dexamethasone (1 μM) and prepared in DMEM LG medium for 16 h to induce insulin resistance [52]. Cells were treated with or without different concentrations of butanol fraction for four hours. Then, the cells were treated with insulin (1 μM) for 30 min and then followed by 2-NBDG for 30 min. The fluorescence intensity of cellular 2-NBDG was measured at an excitation wavelength of 485 nm and an emission wavelength of 535 nm in a fluorescence microplate reader (Spectramax i3x spectrophotometer, Molecular Devices, San Jose, CA, USA). The palmitic acid solution was prepared in conjugated with BSA (1%), as described previously [53].

### 4.7. Differentiation of 3T3-L1 Cells 

Next, 3T3-L1 pre-adipocyte cells were maintained in DMEM supplemented with 10% bovine calf serum in a 6-well plate. The cells were kept at 37 °C in a humidified atmosphere of 5% CO_2_. On Day 0 (two days after 100% confluence), the culture media was replaced by differentiation media (MDI), i.e., DMEM supplemented with 10% FBS, 0.5 mM IBMX, 1 μM Dexamethasone and 10 μg/mL insulin. On Day 2, the media was replaced with 10% FBS-DMEM supplemented with 10 μg/mL insulin. From Day 4 to Day 10, the media were replaced with DMEM containing only 10% FBS, every two days. The non-toxic concentration of BuOJ was treated on Day 0 and Day 2 to evaluate the anti-adipogenic activity as illustrated in Figure 2A. 

### 4.8. Oil Red O (ORO) Staining

On Day 10, the qualitative and quantitative analysis of lipid production was conducted by ORO staining. Cells were washed with 1× PBS (phosphate buffer saline) and then fixed in 10% formaldehyde/PBS solution for 30 min. Cells were washed with 60% isopropanol and the Oil Red O staining solution (6:4, 0.6% ORO dye in isopropanol/water) was added and incubated for 30 min at room temperature. After the staining, cells were washed three times with water. The cells were visualized by bright-field microscopy to evaluate the content of lipids. The quantification of lipid content was carried out by extracting the ORO dyes in isopropanol and measuring at 520 nm using a microplate reader.

### 4.9. Western Blotting

The 3T3-L1 cells were differentiated for 10 days in the presence or absence of BuOJ (60, 80 and 120 µg/mL). On Day 8, cells were collected with the help of a cell scraper. Cell lysis was conducted using ice-cold RIPA buffer containing 25 mM Tris-HCl (pH 7.6), 150 mM NaCl, 1% Nonidet P-40, 1% sodium deoxycholate, 0.1% SDS and a protease inhibitor cocktail (Sigma-Aldrich, St. Louis, MO, USA) for 30 min. Then, the cell suspension (lysates) was separated by centrifugation (14,000 rpm for 20 min at 4 °C). The protein concentration was determined by using a BCA protein assay kit (Pierce, Rockford, IL, USA). Around 30 µg of protein were loaded and separated by 10% SDS-polyacrylamide gel electrophoresis. The proteins in the gels were transferred to the membranes at 150 mA for 1 h and were blocked with PBS containing 5% skim milk and 0.1% Tween 20 for 2 h at room temperature. The blots were incubated with primary and secondary antibodies and the bands were obtained, as mentioned in our previous work [54].

### 4.10. Quantitative Real-Time PCR 

The 3T3-L1 cells were differentiated with or without the treatment of BuOJ (60, 80 and 120 µg/mL). After the completion of the differentiation process on Day 10, cells were collected with the help of a cell scraper and lysed with 1 mL of Trizol. The cell lysate was mixed with chloroform and centrifuged to yield RNA fraction. In the case of liver tissue, after the hominization, RNA was extracted with a QIAzol lysis reagent (Qiagen Sciences, Germantown, MD, USA). The obtained RNA fraction was mixed with 0.5 mL isopropyl alcohol followed by centrifugation (12,000 rpm, 10 min, 4 °C). The precipitated mass of RNA at the bottom of the tube was washed and dried at room temperature for 10 min and it was dissolved in RNase-free diethyl pyrocarbonate (DEPC)-treated water. Then, cDNA was synthesized from the RNA using a cDNA (complementary DNA) synthesis kit (Clontech Advantage^®^ RT-for-PCR kit, #639506). The gene expression levels were analyzed by quantitative real-time (RT) PCR using AB 7900HT Real-Time PCR system (Applied Biosystems #4364346, Foster City, CA, USA). Table 3 shown below illustrates the primers used in the experiments. The initial incubation was for 2 min at 50 °C, and then the cDNA was denatured at 95 °C for 5 min followed by 40 cycles of PCR (95 °C, 20 s, 60 °C, 120 s). The experiment was carried out in triplicate. β-actin as internal control was to normalize the mRNA levels of all genes.

### 4.11. In Vivo Biological Studies (Anti-Obesity Study)

Four-week-old male C57BL/6j mice (average body weight 20 g) were ordered from Central Lab Animal Inc. (Seoul, Republic of Korea). Mice were maintained in accordance with the guidelines for the care and use of laboratory animals by Wonkwang University (Approval No.: WKU19-78). Animal experiments were carried out according to the guidelines prepared by the Institutional Review Board of Wonkwang University. The mouse room was maintained at 25 ± 2 °C and humidity of 55 ± 5% with a 12 h light/12 h dark cycle. The adaptation time given for the mice was one week. Then, the mice were randomly divided into four groups (6 mice in each group), as shown in Table 4. The mice were provided with their respective diets (normal and high-fat diets) and water ad libitum throughout the experiment (14 weeks) [55].

Samples of butanol fraction (suspension prepared in PBS) were administered orally. Normal and control groups were fed with PBS per oral. Body weights and food intake were recorded every week. The food efficiency ratio (FER) was calculated as follows:FER = gained body weight (g) × 100/food intake (g) during the experiment period %

On the last day of the experiment, the mice were induced to fasting for 12 to 15 h, and sacrificed. Blood was withdrawn from the eyes. Blood glucose and triglyceride were determined from the serum. Blood glucose was determined using an ACCU-Check glucose meter (Roche Diagnostics, Mannheim, Germany). Triglycerides (TG) level in serum was determined by using a commercial kit (AM 157S-K, Asan Pharm Co., Ltd., Whasung, Republic of Korea) in accordance with the manufacturer’s instruction. Then, the animals were sacrificed by cervical dislocation. The liver, kidney, spleen and epididymal adipose tissue were removed for the measurement of change in weight and histopathological evaluation.

### 4.12. Histology 

After sacrifice, the mice’s liver and white adipose tissues were immediately fixed in 10% formaldehyde. With alcohol dehydration, the water trapped in the tissue was removed and then embedding was carried out using paraffin. Serial sections of the tissue (5 µm) were cut using a microtome and then mounted on glass microscope slides. Then, the tissue sections were stained with hematoxylin and eosin. The slides were examined using a light microscope and images were captured on a Canon Powershot A640 camera.

### 4.13. Statistical Analysis

Values are expressed as mean ± standard error of the mean. Statistical analysis was carried out by SPSS Statistics 19 software (IBM Co., Armonk, NY, USA) measured by one-way analysis of variance (ANOVA) using Duncan’s multiple range test, Dunnett’s multiple comparisons test and Student’s *t*-test. The values were considered significantly different when *p* < 0.05.

## 5. Conclusions

Overall, this study confirms the anti-adipogenic activity and anti-obesity activity of *Orostachys japonicus* in cell and animal models, respectively. *Orostachys japonicus* was able to inhibit the major (PPARγ and C/EBPα) and minor (SREBP-1c, aP2 and leptin) adipogenic factors. It not only reduced the fat deposition in adipose tissues of mice but also inhibited obesity-induced fibrosis markers in the liver. It has a potent role in diabetes as it has the potential to promote glucose uptake and enhance insulin sensitivity. Thus, it can be concluded that *O. japonicus* has good anti-obesity and antidiabetic activity along with a protective role in the liver against problems generated during obesity.

## Figures and Tables

**Figure 1 pharmaceuticals-17-00357-f001:**
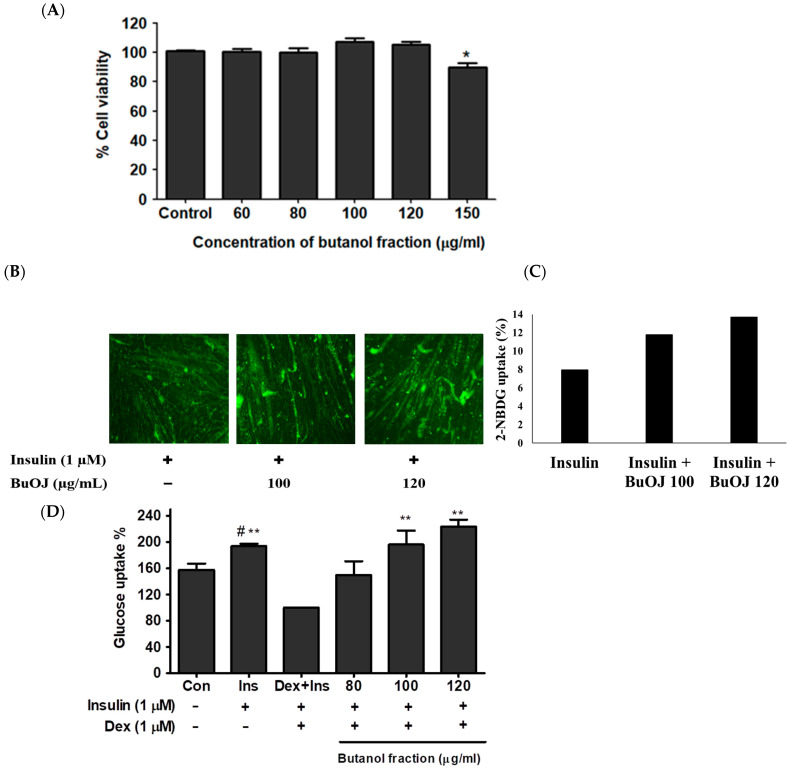
The viability of C2C12 cells in different concentrations of BuOJ was evaluated using the MTT assay. (**A**) Differentiated C2C12 myotubes were treated with insulin alone or along with BuOJ in two different doses. Then, 2-NBDG uptake was assayed in all groups by taking immunofluorescence images with fluorescence microscopy (**B**). The bar diagram (**C**) represents densitometric analysis for the relative fluorescence intensity carried out using Image J software (National Institute of Mental Health, Bethesda, MD, USA) (https://imagej.net/ij/, assessed on 5 February 2022). The C2C12 cells were differentiated and induced insulin resistance using dexamethasone (dex, 1 µM), and the effect of BuOJ on the insulin-medicated glucose uptake was observed in the dexamethasone-induced insulin-resistant C2C12 myoblasts (**D**). The data are expressed as mean ± standard error mean; * *p* < 0.05, ** *p* < 0.01 between the indicated treatment and Dex + Ins, ^#^
*p* < 0.05 between the indicated treatment and control. Statistical significance was determined using Dunnett’s multiple comparisons test.

**Figure 2 pharmaceuticals-17-00357-f002:**
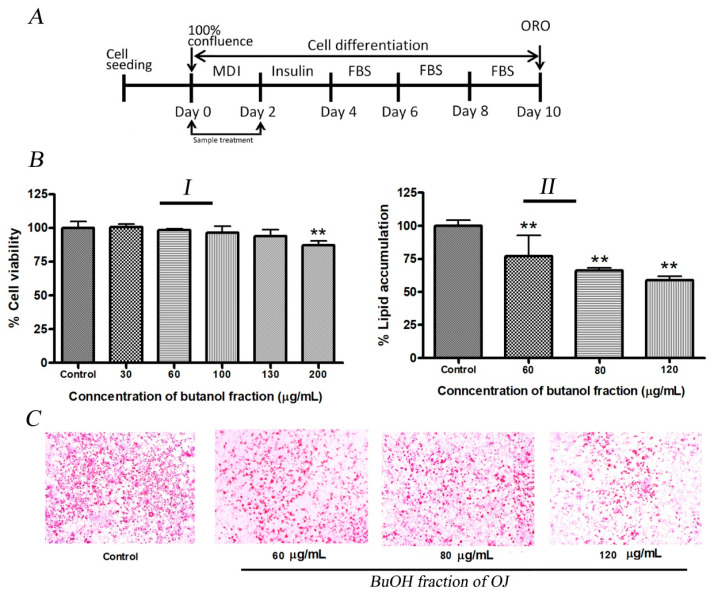
Anti-adipogenic activity of BuOJ on 3T3-L1 cells. (**A**) The 3T3-L1 cells were differentiated with or without the presence of sample (BuOJ) as specified in the illustration. (**BI**) Cell viability study. (**BII**) the non-toxic concentrations 60, 80 and 120 µg/mL were treated on the differentiating 3T3-L1 cells and the lipid production was measured by the ORO study. (**C**) The lipid accumulation in the 3T3-L1 cell differentiated with or without the presence of BuOJ was stained with ORO and the pictures were taken by microscope (**C**). Data shown represent the mean ± SEM from three independent experiments. Statistical significance was determined relative to control by the Student’s *t*-test (** *p* < 0.01).

**Figure 3 pharmaceuticals-17-00357-f003:**
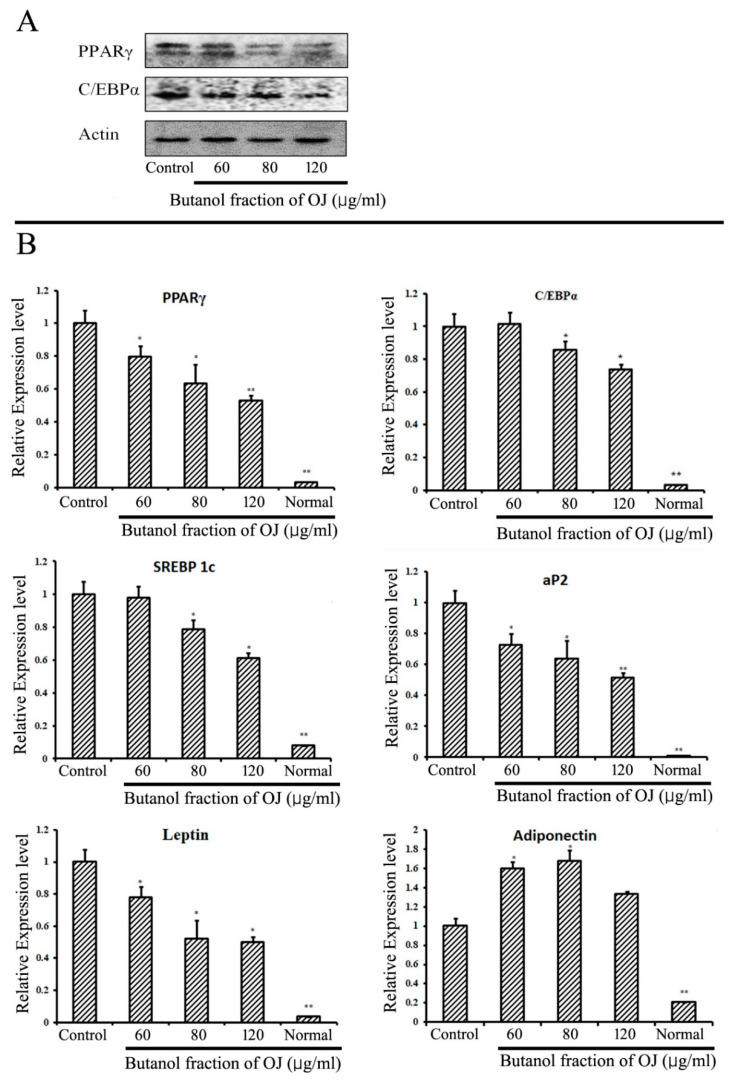
In vitro study of adipogenic markers in 3T3-L1 cells. (**A**) The expression of PPARγ and C/EBPα was carried out by Western blotting analysis in the differentiating 3T3-L1 cells with or without the presence of BuOJ. (**B**) Real-time PCR analysis was carried out to see the expression of PPARγ, C/EBP, SREBP-1c, aP2, adiponectin and leptin in 3T3-L1 cells with or without presence of BuOJ. Data shown represent the mean ± SEM from three independent experiments. Statistical significance was determined relative to control by the Student’s *t*-test (* *p* < 0.05; ** *p* < 0.01).

**Figure 4 pharmaceuticals-17-00357-f004:**
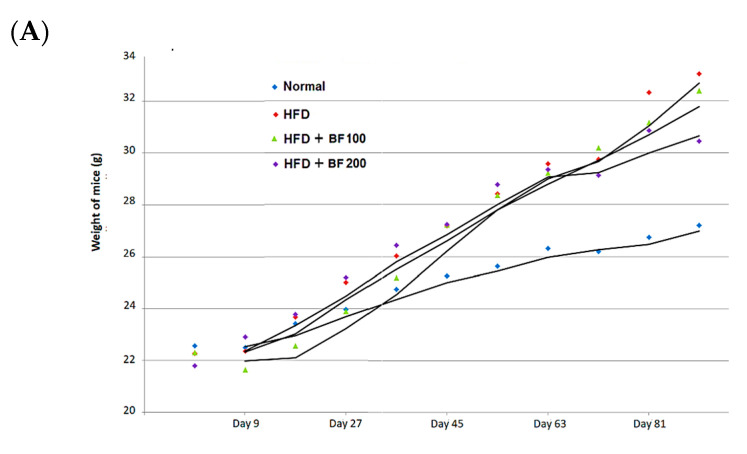
High-fat diet animal experiment for the anti-obesity study. (**A**) Mice were fed with normal diet, HF diet and HF diet supplemented with BuOH fraction (100 and 200 mg/kg) and the growth curve of each group of mice was plotted by calculating the average weight of every nine days measurement. (**B**) After the sacrifice the weight of different organs: liver (**I**), spleen (**II**) and kidney (**III**) were taken for each group of mice and plotted. Data shown represent the mean ± SEM from six independent samples.

**Figure 5 pharmaceuticals-17-00357-f005:**
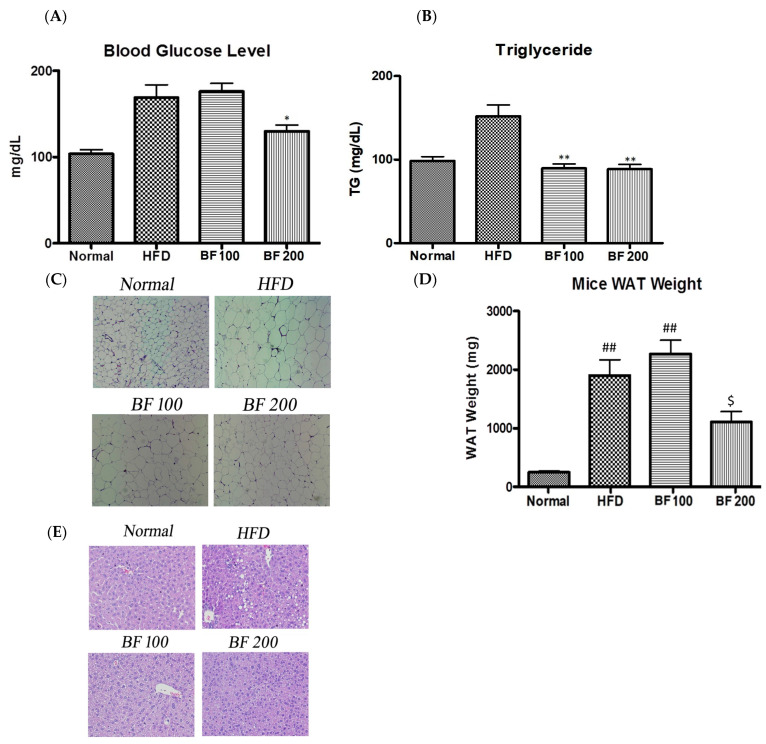
Blood glucose analysis and histology of each group of mice after the sacrifice. (**A**) The blood glucose level. (**B**) The lipid level in the blood of each group of mice evaluated by the measurement of serum triglyceride. (**C**) The histology of adipose tissue. (**E**) The histology of liver tissue. Images (magnification, ×400) were captured to observe the fat accumulation. The average weight of epididymal adipose tissue of each group of mice was evaluated to find the lipid accumulation in adipose tissue (**D**). The data shown are represented as mean ± SEM of six separate samples. Statistical significance was calculated using one-way ANOVA, followed by Dunnett’s multiple comparisons test. * *p* < 0.05 vs. HFD and ** *p* < 0.01 vs. HFD. ^##^
*p* < 0.01 vs. Normal, and ^$^
*p* < 0.05 vs. HFD.

**Figure 6 pharmaceuticals-17-00357-f006:**
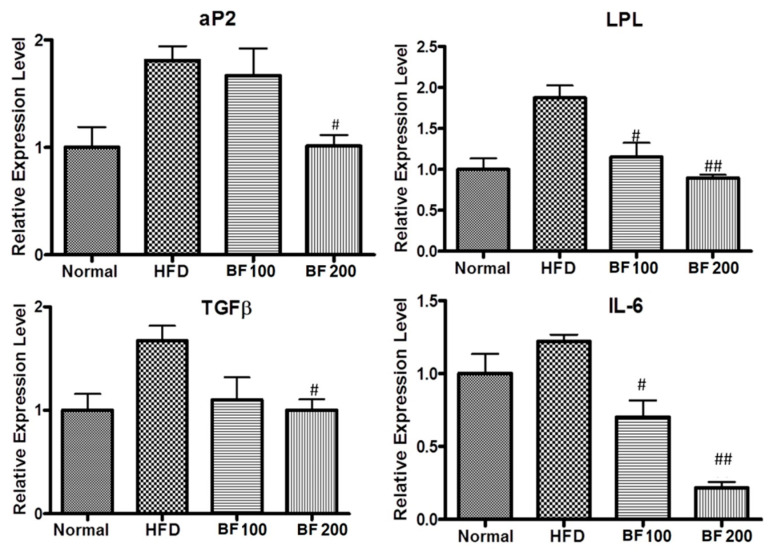
Real-time PCR showing the levels of aP2, IL-6, TGFβ and LPL in the liver samples of different groups of mice. The data shown are represented as mean ± SEM of six separate samples. Statistical significance was calculated using one-way ANOVA, followed by Dunnett’s multiple comparisons test. ^#^
*p* < 0.05 vs. HFD and ^##^
*p* < 0.01 vs. HFD.

**Table 1 pharmaceuticals-17-00357-t001:** IC_50_ values for inhibitory activity against α-glucosidase.

Sample	IC_50_ (µg/mL)
BuOJ	12.13 ± 0.06
Acarbose	132.59 ± 7.14

**Table 2 pharmaceuticals-17-00357-t002:** Body weight, food intake and food efficiency ratio (FER) of different groups of mice.

Group	Initial BW	Final BW	BW Gain	Food Intake (g/day)	FER
Normal	22.56 ± 1.3	27.20 ± 2.1	4.63 ± 0.9 ^a^	5.12 ± 0.2 ^a^	0.90 ± 0.1 ^a^
HFD	22.27 ± 2.1	33.02 ± 1.9	10.74 ± 0.7 ^b^	2.56 ± 0.6 ^b^	4.19 ± 0.9 ^c^
BF 100	22.30 ± 1.5	32.39 ± 1.2	10.08 ± 0.9 ^b^	2.54 ± 0.3 ^b^	3.95 ± 0.7 ^b^
BF 200	21.79 ± 1.8	30.44 ± 2.3	8.651 ± 0.5 ^c^	2.50 ± 0.5 ^b^	3.45 ± 0.2 ^b^

The values are the average of six mice in each group, expressed as mean ± standard deviation. Significant differences are indicated by different letters in a row at *p* < 0.05 as determined by Duncan’s multiple range test.

**Table 3 pharmaceuticals-17-00357-t003:** The primer sequence used for real-time PCR.

Gene	Forward Primer 5′-3′	Reverse Primer 3′-5′
PPARγ	GTG AAG CCC ATC GAG GAC	TGG AGC ACC TTG GCG AAC A
C/EBPα	GCG GGA ACG CAA CAA CAT C	GTC ACT GGT CAA CTC CAG 214 CAC
SREBP-1c	GGT TTT GAA CGA CAT CGA AGA217	CGG GAA GTC ACT GTC TTG GT
Leptin	GCC AGG CTG CCA GAA TTG	CTG CCC CCC AGT TTG ATG
aP2	AGG CTC ATA GCA CCC TCC TGT	CAG GTT CCC ACA AAG GCA TCA C
LPL	TGT ACC AAT CTG GGC TAT GAG ATC AAC	TGC TTG CCA TCC TCA GTC CC
IL-6	GGA AAT CGT GGA AAT GAG	TGC TTG CCA TCC TCA GTC CC
TGFβ	GGA GCA GAG CTG CTG AAA CT	CTT CTC TCC ATC CCT GAC GC
GAPDH	CAA TGA ATA CGG CTA CAG CAA C	AGG GAG ATG CTC AGT GTT GG
β-actin	GTG ACG TTG 220 ACA TCC GTA AAG A	GCC GGA CTC ATC GTA CTC C

**Table 4 pharmaceuticals-17-00357-t004:** Groping of mice with respective diet and sample doses.

Group	Diet	Dose
Normal	Normal diet	Blank PBS
HFD	High-fat diet	Blank PBS
BF 100	High-fat diet	BuOH fraction 100 mg/kg
BF 200	High-fat diet	BuOH fraction 200 mg/kg

## Data Availability

The original contributions presented in the study are included in the articlel, further inquiries can be directed to the corresponding author/s.

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
