# Peer review of "Evaluation of Anti-Obesity and Antidiabetic Activities of Orostachys japonicus in Cell and Animal Models"

_pharmaceuticals, 2024, doi:10.3390/ph17030357_

Round 1

Reviewer 1 Report

Comments and Suggestions for Authors

Why the first table named as Table 3?

L37, effect of O. japonicus should be effect of butanol fraction of O. japonicus.

Why the author investigate Butanol fraction, not CHCl3 fraction? Or use CHCl3 fraction as a comparison? And as best knowledge I know, BuOH fraction not only contain flavonoids, but also alkaloids, terpenoids, etc. How do the author explain the activities of each components?

Can the author use the same symbol for P<0.05, P<0.01? Such as figure 1 use * but table 4 use letters and figure 5 use #. 

The ingredient for normal diet and high fat diet?

What is normal in figure 3b?

Where is SEM for figure 1C and 4A?

In figure caption, please add a brief description as the reader can understand the figure without reading the whole manuscript. And add the full name for the abbreviation. 

Reviewer 2 Report

Comments and Suggestions for Authors

Orostachys japonica is also known as stone pine due to its habit of growing in mountainous areas and its growth habit is similar to the pine cone - a species of flowering plant in the Crassulaceae family native to East Asia. It grows on the surface of mountain cliffs in Korea, Japan and China. The leaves and stems are known to contain several medicinally active components, including fatty acid esters, friedelin, and flavonoids, which have antispasmodic and cytotoxic effects. There are a number of studies where the extract of this plant exhibits antitumor properties.

The review submitted for publication in the journal Pharmaceuticals deserves the highest praise, since the authors were able to painstakingly and carefully collect information about the antidiabetic effect of the plant, conduct experiments on mice, explore the signaling pathways involved, and draw the right conclusions. I believe that the article is worthy of being published in the journal Pharmaceuticals, since the information contained in this work will be very useful for scientists involved in the development of antidiabetic drugs, as well as those working on the topic of obesity, aging and longevity.

Reviewer 3 Report

Comments and Suggestions for Authors

The article presents a study on the medicinal herb Orostachys japonicus, traditionally used in Asian countries. The study aims to establish the plant’s role in lipid and glucose metabolism, positioning it as an anti-obesity and anti-diabetic herb. The research uses a butanol fraction of O. japonicus and evaluates its effects in cell and animal models. In 3T3-L1 cells, lipid inhibition and the expression of adipogenic markers were assessed. The lipid production was evaluated using the oil-red-O technique, while the expression of adipogenic markers was determined through western blotting and RT-PCR. The study also examined the herb’s effect on glucose uptake activity in insulin resistant C2C12 myoblast cells. An animal study was conducted using C57BL mice to evaluate the anti-obesity activity using a high-fat diet model. The researchers assessed serum lipid, blood glucose, adipogenic and fibrosis markers in the liver, and fat deposition in the liver and adipose tissue of the mice.

In conclusion, the study supports previous results and helps establish the potent anti-obesity, anti-diabetic, and liver-protecting effects of O. japonicus. The herb was able to inhibit major and minor adipogenic factors, reduce fat deposition in adipose tissues of mice, inhibit obesity-induced fibrosis markers in the liver, control blood glucose levels, and enhance insulin-sensitivity activity. Thus, O. japonicus demonstrates promising anti-obesity and anti-diabetic activity and a protective role in the liver against obesity-related problems.

Below are some questions that could potentially enhance the effectiveness and readability of the article:

  1. Background Information: Could you provide more information on Orostachys japonicus? What are its traditional uses in Asian countries?
  2. Methodology: Could you elaborate on the methodology used in the study? For instance, how was the butanol fraction of O. japonicus prepared? How were the 3T3-L1 cells and C2C12 myoblast cells cultured and treated?
  3. Results: Could you provide more detailed results? For example, what were the specific changes in the PPARγ, C/EBPα, SREBP-1c, and aP2 expression levels? How significant was the weight, triglycerides, and blood glucose reduction in the obese mice?
  4. Discussion: Could you discuss the implications of these findings? How do they contribute to understanding O. japonicus’s potential as an anti-obesity and anti-diabetic herb?
  5. Limitations and Future Research: What were the limitations of this study, and how might future research address them? Are there other potential medicinal properties of O. japonicus that could be explored in future studies?
  6. Conclusion: Could the conclusion be made more impactful? By summarizing the key findings and their implications concisely and compellingly.

7.      It is advisable to highlight the significance of synthetic endeavors in the advancement of antidiabetic medication. Within this context, it is recommended to underscore the significance of iminosugars and sugar derivatives as potent antidiabetic agents. To support this assertion, referencing the subsequent pertinent articles in the introduction section is suggested: i) https://doi.org/10.1002/anie.202217809 ii) Compain, P.; Martin, O. R. Iminosugars: From synthesis to therapeutic applications; Wiley-VCH:New York, 2007; pp 187−298 and iii) https://doi.org/10.24820/ark.5550190.p011.809.
